# COST-SENSITIVE LEARNING VIA DEEP POLICY ERM

**Lequn Wang, Qiantong Xu, Christopher De Sa, Thorsten Joachims**
Department of Computer Science
Cornell University
Ithaca, NY 14850, USA
`{lw633,qx57,cmd353,tj36}@cornell.edu`

## ABSTRACT

Deep networks for classification are typically trained by maximizing the log likelihood of the training data. However, the conditional probabilities learned in this way are often not well-calibrated and are thus not well-suited for cost-sensitive learning where making different errors incurs different rewards or penalties. In this paper, we propose to directly train neural networks to optimize a cost sensitive loss via Empirical Risk Minimization (ERM). Empirical results show that, with proper initialization, ERM training with cost-sensitive loss outperforms maximum-likelihood training with various form of post-processing on a range of cost-sensitive classification tasks.

## 1 INTRODUCTION

For many real-world prediction problems, different errors incur different costs (Domingos, 1999; Elkan, 2001). For example, consider a bank having to decide whether to provide a loan to an applicant, where mistakenly providing a loan or mistakenly denying a loan clearly has different costs. In principle, training deep networks via maximum likelihood (ML) and classifying via Bayes decision rule can address such cost-sensitive classification problems, but the conditional probabilities learned via ML are known to often be badly calibrated (Guo et al., 2017). In the context of deep networks and beyond, this has led to a number of post-processing approaches aiming to remedy the calibration problem (Platt, 1999; Zadrozny & Elkan, 2001; 2002; Niculescu-Mizil & Caruana, 2005; Naeini et al., 2015; Guo et al., 2017) and the resulting bad cost-sensitive classification performance.

Instead of trying to fix ML training, we propose to directly learn a cost-sensitive decision policy through empirical risk minimization (ERM) (Vapnik, 1998). In this way, we can train any network architecture that ends in a softmax layer as a stochastic policy that takes cost-sensitive actions, instead of using it to model the conditional probability of class membership as in ML training. Our ERM approach directly trains the network policy to minimize cost, thus avoiding the hard problem of learning calibrated probabilities.

We evaluate the proposed cost-sensitive policy learning method on CIFAR-10, CIFAR-100 (Krizhevsky & Hinton, 2009) and the Yelp Review Full (Zhang et al., 2015) dataset. We find that when initialized with ML training in the beginning, directly optimizing a deep network policy to the cost-sensitive loss results in better performance than conventional ML training with post-processing. We note, however, that initialization is important for successful Stochastic Gradient Descent (SGD) training, raising interesting questions around the need for SGD procedures for non-ML objectives.

## 2 COST-SENSITIVE DEEP LEARNING

In the cost-sensitive classification setting, we are given a set of $n$ training examples $S = \{(\mathbf{x}_i, y_i)\}_{i=1}^{n}$ where $\mathbf{x}_i \in \mathcal{X}$ is the input feature vector and $y_i \in \mathcal{Y}$ is the label, as well as a cost matrix $C \in \mathbb{R}^{K \times K}$ where $K$ is the number of classes. Each entry $c_{i,j}$ in the cost matrix $C$ represents the cost of classifying an example that belongs to class $i$ as class $j$. The goal is to learn a classifier such that the expected misclassification cost at prediction time is small.

Works on cost-sensitive learning can be divided into two categories. The first type of work uses probabilities learned by a machine learning model to make cost-sensitive predictions via Bayes

rule (Kukar & Kononenko, 1998; Domingos, 1999). The problem is that the probability estimate is often not accurate especially in deep neural networks. The second category of work extends particular type of classification models to be cost sensitive, such as support vector machines (Tu & Lin, 2010), decision tree (Bradford et al., 1998; Ting, 2002) and neural networks (Kukar & Kononenko, 1998; Zhou & Liu, 2006; Chung et al., 2016). Our work belongs to the second category, but we approach cost-sensitive deep learning from a stochastic policy-learning perspective via ERM.

## 2.1 MAXIMUM-LIKELIHOOD TRAINING WITH POST-PROCESSING

The most common approach to multi-class classification with a deep network $f_w(y|\mathbf{x})$ is to model the conditional probability of label $y$ given feature vector $\mathbf{x}$ using a softmax output layer

$$p_w(y|\mathbf{x}) = \frac{exp(f_w(\mathbf{x}, y))}{\sum_{y' \in \mathcal{Y}} exp(f_w(\mathbf{x}, y'))}. \tag{1}$$

Training is done by maximizing the log likelihood of the training set $S$

$$\hat{w} = \text{argmin}_w \sum_{i=1}^{n} -log(p_w(y_i|\mathbf{x}_i)). \tag{2}$$

If the resulting network $p_{\hat{w}}(y|\mathbf{x})$ succeeds in accurately modeling the conditional probabilities, the optimal cost-sensitive classification rule can be derived via Bayes decision rule

$$\hat{y} = \text{argmin}_{y \in \mathcal{Y}} \sum_{y' \in \mathcal{Y}} p_{\hat{w}}(y'|\mathbf{x}) c_{y,y'}. \tag{3}$$

Unfortunately, it was found that the learned $p_{\hat{w}}(y|\mathbf{x})$ is often not well calibrated, which in turn can lead to bad cost-sensitive classification performance.

## 2.2 COST-SENSITIVE POLICY LEARNING VIA ERM

Instead of modeling the conditional probability, we directly model and train the policy that makes the cost-sensitive classification decisions, thus avoiding the intermediate step of learning the conditional class probabilities. In particular, we reinterpret a deep network $f_w(y|\mathbf{x})$ with a softmax layer as a stochastic policy that selects actions (Joachims et al., 2018)

$$\pi_w(y|\mathbf{x}) = \frac{exp(f_w(\mathbf{x}, y))}{\sum_{y' \in \mathcal{Y}} exp(f_w(\mathbf{x}, y'))}. \tag{4}$$

The training objective is to find a policy $\pi_{\hat{w}}$ that minimizes the cost-sensitive empirical risk

$$\hat{w} = \text{argmin}_w \sum_{i=1}^{n} \sum_{y \in \mathcal{Y}} c_{y_i, y} \pi_w(y|\mathbf{x}_i). \tag{5}$$

A regularizer can be added as well. To pick an action $y$ for a new example $\mathbf{x}$, we can either sample an action from the learned policy $y \sim \pi_{\hat{w}}(Y|\mathbf{x})$, or pick the action corresponding to the mode

$$\hat{y} = \text{argmax}_{y \in \mathcal{Y}} \pi_{\hat{w}}(y|\mathbf{x}). \tag{6}$$

We use mode predictions in the following, because we found that mode predictions perform slightly better than sampling predictions.

## 3 EXPERIMENTS

The following experiments compare the empirical performance of our cost-sensitive policy optimization approach (**CS Policy**) with several variants of ML training with post-processing. In particular, we compare against **ML Max** – predict the class with the maximum value of $p_{\hat{w}}(y|\mathbf{x})$, **ML Bayes** – predict the class that minimizes the expected loss according to Equation (3), and **ML Calibrated Bayes** – first calibrate the model with temperature scaling (Guo et al., 2017) and then predict according to Equation (3). In the following, we present results for image classification and text classification.

Table 1: Cost-sensitive prediction loss on the CIFAR and the Yelp datasets.

| Methods | CS Policy | ML Max | ML Bayes | ML Calibrated Bayes |
|---|---|---|---|---|
| CIFAR-10 | **0.016±0.001** | 0.019±0.001 | 0.019±0.001 | 0.019±0.001 |
| CIFAR-100 | **0.201±0.002** | 0.212±0.002 | 0.210±0.003 | 0.209±0.003 |
| Yelp Review | **0.1090±0.0003** | 0.118±0.006 | 0.115±0.005 | 0.114±0.004 |

Table 2: Const-sensitive prediction loss for different intermediate losses on CIFAR10.

| Intermediate Loss $i$ | 0 | 0.2 | 0.4 | 0.6 | 0.8 | 1 |
|---|---|---|---|---|---|---|
| ML Max | 0.0132 | 0.0262 | 0.0391 | 0.0521 | 0.065 | 0.0780 |
| ML Bayes | 0.0127 | 0.0259 | 0.0387 | 0.0518 | 0.0651 | 0.0780 |
| ML Calibrated Bayes | 0.0119 | 0.0252 | 0.0383 | 0.0517 | 0.0647 | 0.0780 |
| CS Policy (ML init) | **0.0117** | **0.0227** | **0.0366** | **0.0490** | **0.0596** | 0.0756 |
| CS Policy (random init) | 0.0156 | 0.1026 | 0.1505 | 0.1025 | 0.0673 | **0.0728** |

For image classification, we perform experiments on the CIFAR-10 and CIFAR-100 (Krizhevsky & Hinton, 2009) datasets, which consist of images of 10 classes in 2 super classes and 100 classes in 20 superclasses respectively. We define the cost sensitive loss as: (1) If an example is classified correctly, then the loss is 0. (2) If an example is misclassified but in the same super class, the loss is a constant intermediate loss $i \in [0, 1]$. (3) If an example is misclassified into another super class, then the loss is 1. For text classification, we perform experiments on the Yelp Review Full dataset (Zhang et al., 2015). The dataset consists of reviews of 5 classes which are the number of stars users rated the items. The cost sensitive loss of classifying a review of star $a$ as star $b$ is defined as $0.25 * |a - b|$. Throughout our experiments, we randomly select 10% of the training dataset as validation set and the rest as the new training set for all the three datasets.

We use the ResNet20 architecture (He et al., 2016) for image classification and the Very Deep Convolutional Neural Network with MaxPooling (Conneau et al., 2016) for text classification. All of the experiments are trained with regular SGD with momentum parameter 0.9 (Sutskever et al., 2013). For image classification, the models are trained for 1000 epochs and the learning rates are decayed by factors of 10 at 500 and 750 epochs. For text classification, models are trained for 12 epochs with the learning rate schedule from (Conneau et al., 2016). For ML training, we grid-search the learning rate in {0.001,0.002,0.005,0.01,0.02,0.05,0.1,0.2,0.5} and {0.001,0.002,0.005,0.01,0.02,0.05} and weight decay in {1e-5,2e-5,5e-5,1e-4,2e-4,5e-4} and {1e-6,2e-6,5e-6,1e-5,2e-5,5e-5} for image classification and text classification respectively. For CS Policy training with ML initialization, we initialize the policy networks with ML training (hyperparameters selected for ML training) for half of the epochs, and then train using the ERM objective from that starting point. We also grid-search the learning rate and weight decay in the same range. For CS Policy with random initialization, we train the policy from scratch, again grid-searching the hyperparameters as above. Experiments with the final hyperparameters are repeated 5 times to assess statistical significance.

The results for image classification with intermediate loss $i = 0.1$ and text classification are given in Table 1. The results show that training with cost-sensitive loss significantly outperforms models trained on log loss even when they are post-calibrated. More details about the relative performance of the methods is given in Table 2 for the CIFAR10 dataset for different intermediate losses. First, it shows that initialization of the policy-learning objective is very important. Second, the policy optimization approach with such initialization gives superior cost-sensitive classification performance over a range of intermediate losses $i$.

## 4  DISCUSSION

In this paper we proposed to optimize the cost-sensitive classification performance of deep networks via policy learning and ERM. We found that with good initialization, ERM training with cost sensitive loss leads to performance gains, avoiding the difficult problem of well-calibrated conditional probability estimation. Future work includes improved SGD algorithms that are less reliant on initialization, and extending full-information cost-sensitive learning to conterfactual cost-sensitive learning (Joachims et al., 2018). This work was supported in part by NSF award IIS-1615706.

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
