# OpenReview forum: "Cost-Sensitive Learning via Deep Policy ERM"
_ICLR.cc/2018/Workshop — Reject_

### Official Review · AnonReviewer3 · 2018-03-08
**Simple but effective idea**

**Rating:** 6
**Confidence:** 3

**Review:**

In the context that there is a cost of misclassifying an example, there is usually mis-calibration issues for the conditional probability. The manuscript proposes to directly optimize the cost-sensitive objective and shows that the proposed approach outperforms post-processing approach with proper tuning, and thus avoid the hassle of having an additional step of post-processing calibration.

The results seem promising. But it seems that the proposed approach heavily relied on initialization. If trained from scratch, the performance would be much worse. The result of proposed approach in Table 2 is way worse when intermediate loss is from 0.2 to 0.6. The CS Policy (ML init), from my perspective, is another post processing approach, where in the first half epoch the model is trained with normal objective and then fined-tuned in the second half. Having said that, the results are better than other baselines.

---

### Official Review · AnonReviewer2 · 2018-03-10
**limited originality. Experiments only provide partial information.**

**Rating:** 3
**Confidence:** 4

**Review:**

The paper proposes a method for dealing with class sensitive classification. It consists in optimizing the Bayes risk for cost sensitive classification and then using the argmax decision rule on the classifier predictions. The classifier output is a softmax. Experiments are performed on image and text classification tasks. Comparisons are performed with ML trained baselines.

This seems a trivial idea, probably already used several times. This is similar for example to one of the strategies propose in ref. (Kukar et al 1998), but with a different training loss (they used a cost sensitive MSE). Concerning the performance, you mention the cost sensitive loss, but not the classical (0/1 loss) accuracy. Your decision rule being not the cost sensitive Bayes rule, but the 0/1 loss Bayes rule, you should also indicate the corresponding score. The training and decision criteria are different, why? You should also provide the performance with the cost sensitive Bayes decision rule. Also, the title is misleading since in this paper you do not use any policy but a simple argmax decision rule.

---

### Official Review · AnonReviewer1 · 2018-03-12
**good results but rather preliminary**

**Rating:** 5
**Confidence:** 4

**Review:**

The paper interprets a soft-max output deep network as a policy for cost-sensitive learning, and then use deep policy ERM to train a good network for a given cost matrix. Experimental results show that the deep policy ERM achieves better performance than probability calibration models.

Pros:
  * simple idea that empirically works
  * clear illustration

Cons:
  * not comparing with state-of-the-art cost-sensitive deep learning models in Chung et al., 2016; in fact, Chung et al. highlighted the importance of initialization, which echoes the authors' findings in this paper, but the relationship is not carefully discussed
  * not comparing with standard cost-sensitive benchmarks like those in Domingos, 1999
  * the results are good but not surprising in the cost-sensitive literature, as many existing works shows that models trained with cost information can perform better than models trained without the cost information. So the results do not bring much more insights.

---

### Decision · Program_Chairs · 2018-03-20
**ICLR 2018 Workshop Acceptance Decision**

**Decision:**

Reject

**Comment:**

Based on the reviews, this paper has not been accepted for presentation at the ICLR workshop. However, the conversation and updates can continue to appear here on OpenReview.